# Numerical simulation of rock blasting under different in-situ stresses and joint conditions

**Hai Rong, Nannan Li◉\*, Chen Cao, Yadi Wang, Jincheng Li, Mingda Li**

College of Mining, Liaoning Technical University, Fuxin, China

\* 934744832@qq.com

**Data Availability Statement:** All relevant data are included within the paper.

**Funding:** This work was supported by the National Natural Science Foundation of China Project No. 51904145, Basic scientific research project (youth project) of Liaoning Provincial Department of

## Abstract

High primary rock stress can limit the generation of rock cracks caused by blasting, and blasting usually shows different rock breaking states under different primary rock stress conditions. There are a large number of naturally formed joints in rock mass, due to the limitations of laboratory tests, a numerical model of jointed rock mass was established using LS-DYNA software to investigate the evolution of blasting damage under various in-situ stresses and open joints. In this simulation, using the Lagrange-Euler (ALE) procedure and the equation of state (JWL) that defines explosive materials, the study considered different joint thicknesses (2cm, 4cm, and 6cm), joint angles (0˚, 30˚, 60˚, and 90˚), and in-situ stress conditions (lateral stress coefficients of 0.5, 1, and 2, with vertical in-situ stresses of 10MPa and 20MPa), through stress analysis and damage area comparison, the relationship between damage crack propagation and horizontal and vertical stress difference is explored. The research aimed to understand the mechanisms underlying crack initiation and propagation. The results show that: (1) The presence of joints exerts a barrier effect on the expansion and penetration of cracks. When explosion stress waves reach the joint surface, their propagation is impeded, leading to the diffusion of wing cracks at the joint ends. When the lateral stress coefficient and joint angle are the same, an increase in initial in-situ stress results in a reduction in the area of the blasting damage zone. (2) Under the same initial in-situ stress conditions, the area of the blasting damage zone initially increases and then decreases with an increasing joint angle. However, it remains larger than that without a joint, and there exists an optimal angle that maximizes the damage area. In the simulated conditions, the area of damage cracks is greatest when the joint angle is 60˚ dip angle. (3) The presence of initial in-situ stress has a certain impact on the initiation and expansion of blasting cracks. The degree and nature of this influence are not solely related to the lateral stress coefficient but also depend on the joint's angle and thickness. When in-situ stress is present, the initial in-situ stress field's pressure is not conducive to the initiation and propagation of blasting cracks. However, the existence of a joint has a noticeable guiding and promoting effect on crack propagation, and the pattern of crack propagation is influenced by both joint and in-situ stress conditions.

Education in 2022 No. LJKQZ20222322, the
Engineering Laboratory of Deep Mine Rockburst
Disaster Assessment Open Project No.
LMYK2020006, the Liaoning Natural Science
Foundation Program Guidance Plan No. 2019-ZD-
0045, and the Liaoning Provincial Department of
Education Project No. LJ2019JL007.

**Competing interests:** The authors have declared
that no competing interests exist.

## 1 Introduction

Mesomorphic rock mass is widely distributed in nature, accounting for about 66.7% of the
land area, and 77.3% in China [1]. The horizontal stratified rock mass has the characteristics of
transverse isotropy, and there are structural weak planes inside the rock mass. The composi-
tion of the rock mass is basically the same parallel to the structural plane, while the direction
perpendicular to the structural plane shows frequent alternations of soft and hard. At the same
time, there are many joints and cracks in natural rock mass, resulting in obvious anisotropy in
the aspects of force and deformation [2–5].

Currently, blasting remains the primary method for rock fragmentation. In the process of
rock breaking, as the propagation distance increases, the shock wave generated by explosive
detonation rapidly transforms into a stress wave [6], playing a crucial role in the rock fragmen-
tation process. The presence of naturally occurring joints, cracks, bedding, faults, and other
structural features within natural rock masses results in variations in the mechanical proper-
ties, vibrations, permeability, and energy transfer characteristics of these rock formations [7–
9]. The propagation and attenuation of explosion-induced stress waves within such rock mas-
ses, which contain joints, cracks, and faults, are also subject to alteration due to these structural
features, ultimately impacting the effectiveness and safety of engineering blasting operations
[10–12]. Therefore, it is of great significance to study the law of explosive crack propagation
and stress wave propagation of jointed rock mass under explosion load to improve the blasting
energy utilization efficiency, rock breaking effect and safety of rock mass engineering [13].
Domestic and foreign scholars have conducted in-depth studies on this. Chai Shaobo et al.
[14] compared the propagation theory of explosion stress waves in rock mass with cross-
jointed joints with numerical models, and further discussed the propagation law of explosion
stress waves in rock mass with cross-jointed joints. Wang Shumin et al. [15] introduced the
Poyting-Thomson model as the discontinuity condition and derived the propagation equation
of stress waves through a set of parallel viscoelastic joints based on the time-domain recursion
method to explore the influence of viscoelastic joints on the propagation of stress waves in
rock mass. Xu Bangshu et al. [16] studied the blasting parameters of horizontal layered rock
mass with large section where joint fissured development, carried out field blasting tests and
the failure mechanism analysis of layered rock mass, optimized the smooth blasting parame-
ters of tunnel excavation, cut hole layout scheme and maximum single-hole charge parameter,
the characteristics of surrounding rock, overcut and undercut of tunnel contour and deforma-
tion of surrounding rock after explosion are compared and analyzed. Niktabar et al. [17] used
a large direct shear testing machine to study the shear properties of jointed rock mass in differ-
ent roughness seasons, aiming at the fact that the jointed rock mass is often subjected to
dynamic loading by blasting during mining. The results show that the shear strength of regular
joints is higher than that of ordinary joints, and the shear strength of rock mass decreases with
the increase of shear times. Roy et al. [18] conducted experimental studies on fracture failure
modes of jointed rock masses with different fracture toughness and tensile strength, and the
results showed that: With the decrease of joint thickness, the fracture toughness and tensile
strength of rock mass decrease, and the fracture process zone of jointed rock mass is less sensi-
tive to the direction of the joint, which is related to the thickness of the joint. The interaction
of uneven surfaces at the joint increases the friction resistance and dissipates the fracture
energy of the sample. Yang et al. [19] studied the blasting crack propagation characteristics of
jointed rock mass under high stress conditions by using a digital laser dynamic caustic line
experiment system, obtained the crack initiation mode, stress intensity factor and crack veloc-
ity at the joint, and concluded that the state of high stress significantly increased the shear fail-
ure degree of crack initiation at the joint, resulting in an increase in the crack initiation Angle.

Miranda et al. [20] analyzed the influence of joint geometry and number on acoustic wave propagation through acoustic wave test. The test results of Singh and Sastry [21] show that the average block degree of blasting is controlled by the intersection Angle between the structural plane and the free plane. In the range of 0° to 90°, the average block degree increases with the increase of the intersection Angle. Michal Kucewicz et al. [22] proposed a calculation method of KCC constitutive model and a new strategy based on optimization to effectively calibrate brittle damage parameters. The fracture energy and fracture toughness were determined by experimental tests. The comparison shows that the method can improve the efficiency of fracture reproduction. Pawel Baranowski et al. [23] studied the damage of dolomite through small-scale blasting test. The results show that the heterogeneity and initial cracks have significant effects on the observed failure and cracking patterns. Comparisons of acceleration histories, scabbing failure, and number of radial cracks and crack density confirmed the overall repeatability of the actual testing data.

Some scholars have studied the influence of joint on explosive crack propagation and stress wave propagation in rock mass by numerical simulation. For example, Xie Bing et al. [24] and Qu Shijie et al. [25] respectively simulated and analyzed the influence of joint geometric parameters on the pre-cracking joint formation effect, and concluded that the greater the Angle of joint group and gun hole connection, the better the joint formation effect. Under the same conditions, the smaller the joint Angle and spacing, the more difficult it is to form connected cracks, and the easier it is to form sawtooth cracks. Wei Chenhui et al. [26] and Zhang Fengpeng et al. [27] respectively studied the propagation of blasting cracks in rock mass under different initial stresses and joint characteristic parameters, and the results showed that the reflection tensile failure of the joint surface was weaker than that without filling. The coupling between the tensile crack generated by the reflection tensile action of the joint surface and the blasting main crack enhanced the fracture degree of the rock mass between the explosion source and the joint. The existence of the joint is conducive to the propagation of the explosive crack along the joint plane. Deng et al. [28] used DEM method to conduct numerical simulation on the damage of circular tunnel under the action of explosion shock wave, and studied the spatial mechanical characteristics of jointed rock mass, the initial stress of rock mass and the impact of shock wave amplitude on tunnel damage. The results show that the joint direction in the rock mass surrounding the tunnel has a great influence on tunnel damage. The initial stress of the tunnel has little influence on the damage of the tunnel, and the bolting support can greatly improve the stability of the tunnel. Pawel Baranowski et al. [29] obtained the parameters of damage and fracture based on the laboratory support method of weight reduction impact and numerical simulation, and introduced the method of determining the parameters of dolomite JH-2 model. Michal Kucewicz et al. [30] studied Johnson-Holmquist II (JH-2) model, Johnson-Holmquist concrete (JHC) model and Karagozian and Case concrete (KCC) model. Through numerical simulation, their performance under different stress conditions is evaluated, and their effectiveness in reproducing dolomite behavior is verified, and the effectiveness of JH-2 model in simulating the drilling and blasting process of working face is proved. Pawel Baranowski et al. [31] proposed a multi-scale modeling and simulation method for rock mass destruction blasting, which laid the foundation for the initial conditions of the global three-dimensional finite element model. The effectiveness of the method was verified by comparing the simulation analysis with the experimental results.

In summary, existing research in the field of the blasting mechanism of jointed rock masses has primarily focused on experimental investigations. While there have been extensive discussions in numerical simulations regarding joint parameters such as thickness, angle, numerical control, filling strength, and length, a noticeable gap remains in the study of joint parameters specifically for double-hole single joints under various conditions. Building upon the work of

previous researchers, this paper aims to further explore the patterns of crack propagation and the propagation of explosive stress waves during the blasting process of jointed rock masses. Additionally, it takes into account different in-situ stress conditions and conducts numerical simulations to study the influence of in-situ stress conditions, joint thickness, and joint angle (the angle between the joint plane and the hole axis) on the initiation and propagation of explosive cracks.

## 2. Establishment of numerical model

### 2.1 Calculation model

In this study, a computational model incorporating joint surfaces was established using the finite element software LS-DYNA. The specific model is illustrated in Fig 1, with dimensions of 700cm×600cm. Two circular holes with a diameter of 100cm were excavated within the model, and the spacing between the holes was set at 300cm, and the open (empty) joint is set between the two holes (the distance between the holes is adjusted to 500 cm to keep the joint length unchanged considering that the Angle of the joint is 0˚). Non-reflective boundary conditions were applied to the perimeter of the model, with dimensions given in g-cm-us units. To better illustrate the impact of the joint on explosive stress waves, monitoring points were strategically positioned within the computational model. Monitoring points were selected at the joint surface, with one monitoring point designated for every 10 units along the joint. The model employed SOLID 164 hexahedron units, the ALE algorithm was utilized for modeling explosives and air, and a coupled analysis was conducted between explosives, air, and the rock mass.

### 2.2 Explosive and air parameters and equation of state

The high performance explosive material model *MAT_HIGH_EXPLOSIVE_BURN built in ANSYS/LS-DYNA software and the JWL equation of state are used to describe the volume, pressure and energy characteristics of the explosive products during the explosion. The expression is as follows:

$$P = A\left(1 - \frac{\omega}{R_1 V}\right)\exp(-R_1 V) + B\left(1 - \frac{\omega}{R_2 V}\right)\exp(-R_2 V) + \frac{\omega E_0}{V}$$

Where: $P$ is the detonation pressure; $V$ is the relative volume; $E_0$ is the initial specific internal energy; $A$, $B$, $R_1$, $R_2$, $\omega$ are material constants, as shown in Table 1.

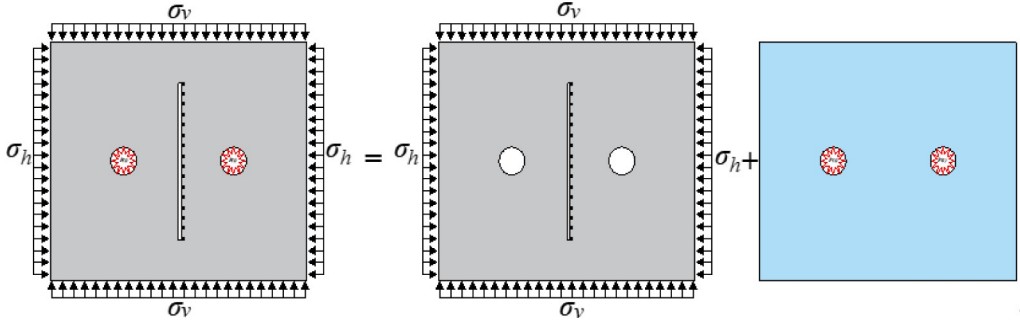

**Fig 1. Numerical model.**

**Table 1. The parameters of explosive.**

| $\rho_e/(kg/m^3)$ | $V_0D(m/s)$ | $P_{CJ}/GPa$ | A/GPa | B/GPa | $R_1$ | $R_2$ | $\omega$ | $E_0/GPa$ |
|---|---|---|---|---|---|---|---|---|
| 1320 | 6690 | 16 | 586 | 21.6 | 5.81 | 1.77 | 0.282 | 7.38 |

Air adopts the empty matter material model *MAT_NULL, and its equation of state is expressed in linear polynomial *EOS_LINER_POLYNOMIAL:

$$P = C_0 + C_1 V + C_2 V^2 + C_3 V^3 + (C_4 + C_5 V + C_6 V^2)E_0$$

In the formula, $C_0 \sim C_6$ are the relevant parameters of the equation. Among them, $C_4 = C_5 = 0.4$, $E_0 = 2500 MJ/m3$, $V = 1.0$, and other parameters are 0.

### 2.3 Rock parameters

The geometric model involves three distinct materials: granite, explosives, and air. The Arbitrary-Lagrange-Euler method in LS-DYNA was applied in this simulation. This method can allow nonlinear computational convergence and computational efficiency. The former material was set as Lagrange parts while explosive and air were set as Euler parts. Rock materials are characterized as porous and brittle, featuring numerous pores and micro-cracks. When subjected to external forces, these cracks undergo progressive development and interpenetration, culminating in the formation of extensive macroscopic fractures, which ultimately lead to material damage and failure. The fracture development process in rock materials consists of several distinct stages, including elasticity mechanics, fracture mechanics, and damage mechanics [32]. RHT model is a tensile and compressive damage model proposed by Riedel, Hiermaier and Thoma et al. [33]. Based on the HJC model, which considers the impact of the failure strength of rock materials under blasting and dynamic loads on the impact pressure, strain rate, strain hardening and damage softening of rock materials. The study of WANG et al. [11] shows that RHT model has good applicability to the blasting simulation of rock materials. In this paper, RHT model is used for numerical simulation research, and the specific parameters [33] are shown in Table 2.

## 3 Simulation results and analysis

### 3.1 The influence of temporal thickness on crack propagation under geostress is not considered

Fig 2 illustrates the comprehensive development of peripheral cracks with different joint widths when subjected to explosive stress waves and explosive gas pressure, without taking into consideration the effects of in-situ stress ($\sigma_v = \sigma_h = 0$). As depicted in Fig 2(A), in the absence of any joints, an initial stage of the explosion leads to the formation of a small crushing zone near the two gun holes. Beyond this crushing zone, a radial crack zone emerges with random cracking patterns. The propagation of stress waves to the depth of the gun holes causes the gradual expansion of radial cracks, resulting in the formation of several primary and secondary cracks (350us). The crushing zone arises due to the radial compressive stress exceeding the compressive strength of the rock, while the radial cracks stem from the circumferential tensile stress surpassing the tensile strength of the rock. As the stress wave travels and its energy gradually dissipates during the propagation process, the cracking effect weakens as well. At 350us, the stress wave intersects with the two gun holes. At this point, the crack propagation speed is slower than the stress wave propagation speed, and the crack between the gun holes has not fully penetrated. Subsequently, under the influence of explosive gas, the radial primary

**Table 2. RHT constitutive model parameters for rock [34].**

| Parameter | Value | Parameter | Value |
|---|---|---|---|
| Mass density (kg/m$^3$) | 2660 | Break compressive strain rate | 3E+25 |
| Elastic shear modulus (GPa) | 21.9 | Break tensile strain rate | 3E+25 |
| Relative shear strength | 0.18 | Lode angle dependence factor Q0 | 0.68 |
| Relative tensile strength | 0.04 | Lode angle dependence factor B | 0.01 |
| Parameter for polynomial EOS T1 (GPa) | 35.27 | Compressive yield surface parameter | 0.53 |
| Parameter for polynomial EOS T2 (GPa) | 0 | Tensile yield surface parameter | 0.7 |
| Damage parameter D1 | 0.04 | Crush pressure (MPa) | 125 |
| Damage parameter D2 | 1.0 | Compaction pressure (GPa) | 6 |
| Hugoniot polynomial coefficient A1 (GPa) | 35.27 | Shear modulus reduction factor | 0.5 |
| Hugoniot polynomial coefficient A2 (GPa) | 39.58 | Eroding plastic strain | 2.0 |
| Hugoniot polynomial coefficient A3 (GPa) | 9.04 | Minimum damaged residual | 0.01 |
| Failure surface parameter A | 1.60 | Porosity exponent | 3.0 |
| Failure surface parameter N | 0.61 | Initial porosity | 1.0 |
| Residual surface parameter AF | 1.60 | Pressure influence on plastic flow in tension | 0.001 |
| Residual surface parameter NF | 0.61 | Tensile strain rate dependence exponent | 0.036 |
| Parameter for polynomial EOS B0 | 1.22 | Compressive strength (MPa) | 167.8 |
| Parameter for polynomial EOS B1 | 1.22 | Compressive strain rate dependence exponent | 0.032 |
| Reference compressive strain rate | 3E-05 | Gruneisen gamma | 0 |
| Reference tensile strain rate | 3E-06 | | |

crack is further extended (550us, 750us), eventually resulting in the connection of the explosive crack between the two holes (1000us). This numerical simulation effectively recreates the entire process of the formation of a crushing zone, a fissure zone, and the interpenetration of cracks between the gun holes within the surrounding rock mass when subjected to blasting.

As depicted in Fig 2(B)–2(D), when a vertical open joint is present between the two gun holes, and the joint distance is equivalent to that between the two gun holes, a similar scenario to Fig 2(A) unfolds under the influence of explosive stress waves, 350us earlier. In this case, the presence of the joint has a relatively minor impact on crack propagation. At 350us, in the absence of any joint, the stress wave meets and superimposes, elevating the stress at the center of the connection between the two holes. However, no rock precracking is observed at the center. With the presence of joints, it becomes challenging for stress waves to traverse through the joints. The existence of open joints leads to the reflection of compressive stress waves into tensile stress waves, resulting in the formation of a tensile failure zone at the joint plane. Rock near the joint plane experiences damage from both compressive and tensile stress waves. Cracks initiate at the center of the joints and compress the joints, causing the two joint surfaces to meet and then subjecting them to tensile forces. This process involves partial intersection and separation of the joint surfaces. From 350us to 1000us, the cracks continued to expand under the influence of explosive stress waves and explosive gas. After 550us, the rate of crack expansion decreased. At this point, the explosive stress wave had propagated to the end of the joint, and the distance between the joints shortened until the joint surfaces came into contact with each other.

As illustrated in Fig 3, stress peaks are extracted based on the selected elements. The stress variation pattern observed in the figure is approximately "W"-shaped. The stress profiles at the two ends of the joint almost mirror each other and align with the other two curves. When the joint thickness is 2cm, the maximum stress peak at the center is 216.2MPa, while the minimum stress is 51.9MPa. With a joint thickness of 4cm, the maximum stress at both ends of the joint

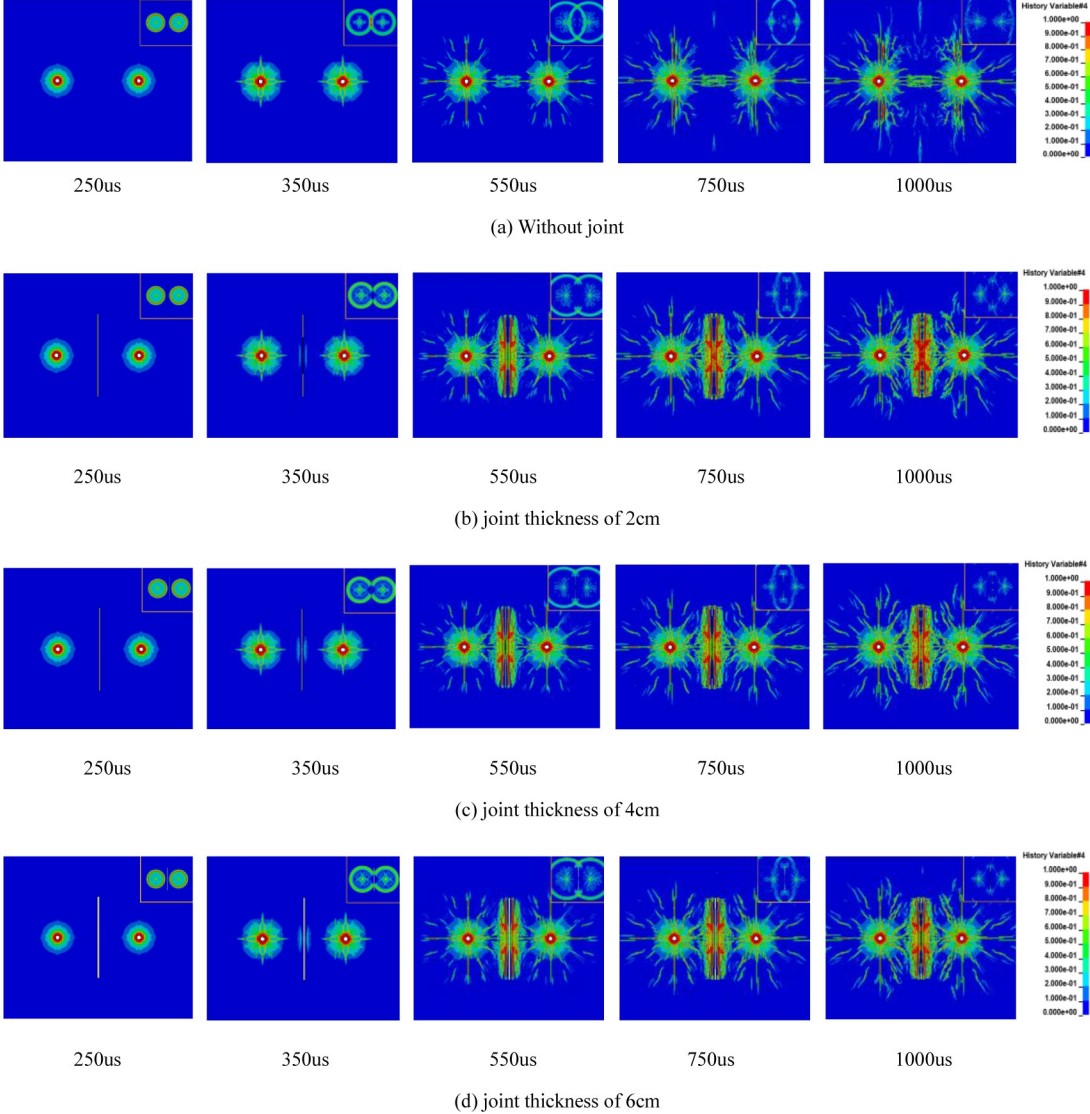

**Fig 2. Evolution of explosive cracks in rock mass with different joint thicknesses without in-situ stress (stress wave cloud image at upper right corner).**

is 193.7MPa, and the minimum stress is 16MPa. For a joint thickness of 6cm, the maximum stress at both ends of the joint is 203.1MPa, and the minimum stress is 17.3MPa. It's worth noting that the stress at both ends of the joint exceeds the compressive strength of the rock, resulting in rock failure, and the rock between the two ends predominantly experiences tensile stress.

In Fig 4, it's evident that the displacements at both ends of the joint remain relatively constant, irrespective of the joint thickness, and exhibit a symmetrical relationship. When the joint thickness is 2cm and 4cm, the displacement of the element converges within a horizontal range. This occurs because the small joint thickness leaves no space for movement between the joint surfaces upon contact. The maximum displacement is 1.04cm and 2.04cm, respectively. Conversely, when the joint thickness is 6cm, there's no contact between the two joint surfaces, and the maximum displacement is 2.44cm.

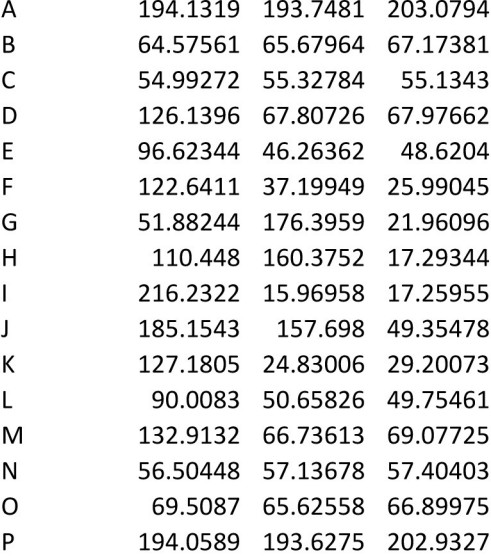

| | | | |
|---|---|---|---|
| A | 194.1319 | 193.7481 | 203.0794 |
| B | 64.57561 | 65.67964 | 67.17381 |
| C | 54.99272 | 55.32784 | 55.1343 |
| D | 126.1396 | 67.80726 | 67.97662 |
| E | 96.62344 | 46.26362 | 48.6204 |
| F | 122.6411 | 37.19949 | 25.99045 |
| G | 51.88244 | 176.3959 | 21.96096 |
| H | 110.448 | 160.3752 | 17.29344 |
| I | 216.2322 | 15.96958 | 17.25955 |
| J | 185.1543 | 157.698 | 49.35478 |
| K | 127.1805 | 24.83006 | 29.20073 |
| L | 90.0083 | 50.65826 | 49.75461 |
| M | 132.9132 | 66.73613 | 69.07725 |
| N | 56.50448 | 57.13678 | 57.40403 |
| O | 69.5087 | 65.62558 | 66.89975 |
| P | 194.0589 | 193.6275 | 202.9327 |

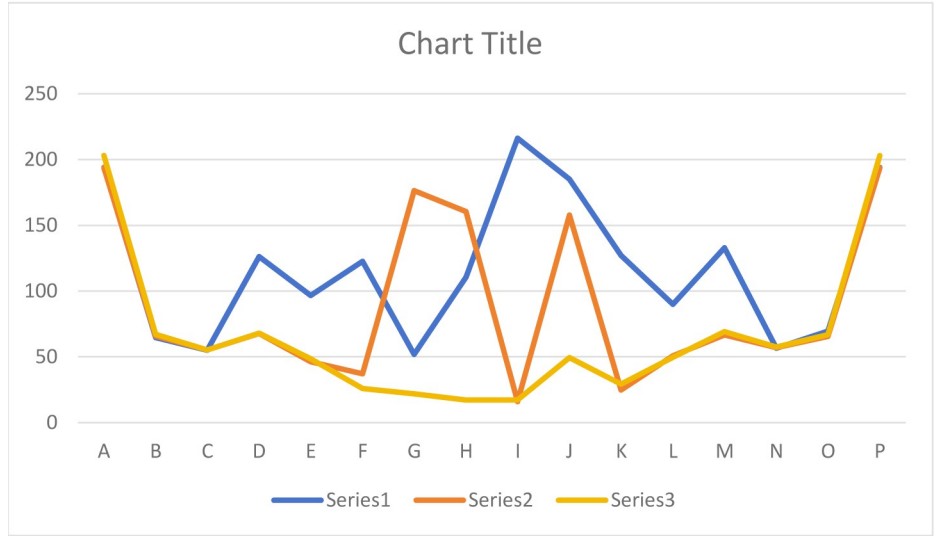

**Fig 3. Peak stress curve of each unit.**

## 3.2 The effect of geological stress on crack propagation is not considered

As depicted in Fig 5(A), when there is a horizontal joint between the two holes, and the joint is not connected to the two holes, crack propagation at 250us under the influence of the explosive stress wave resembles the 250us scenario in Figs 1 and 5(C) and 5(D). It results in the formation of a confined crushed zone and a radial micro-crack zone. The crack propagation tends to incline towards the end of the joint. With the passage of time, the existing radial micro-cracks continue to spread further in their original direction. At 550us, horizontal cracks connect with the joint. As the cracks expand, both main and secondary cracks gradually form. The cracks near both sides of the joint plane continue to widen under the influence of the explosive gas, eventually breaking through. The number of cracks between the two holes exceeds those in the opposite direction and surrounds both sides of the joint plane. It's evident in the stress wave cloud diagram that, as the stress wave propagates to the joint's end, local damage occurs at the joint end. The presence of the joint alters the stress wave's propagation path. The stress wave

| | | | |
|---|---|---|---|
| A | 0.0426 | 0.04959 | 0.05102 |
| B | 0.76337 | 0.76944 | 0.78316 |
| C | 1.00399 | 1.30673 | 1.30732 |
| D | 1.03911 | 1.65517 | 1.67257 |
| E | 1.03114 | 1.77489 | 1.79622 |
| F | 1.04076 | 2.0224 | 2.09862 |
| G | 1.03987 | 2.04343 | 2.3023 |
| H | 1.03973 | 2.04223 | 2.44118 |
| I | 1.04324 | 2.04135 | 2.41706 |
| J | 1.03896 | 2.04159 | 2.28143 |
| K | 1.03849 | 2.00965 | 2.06245 |
| L | 1.0319 | 1.75892 | 1.78271 |
| M | 1.04091 | 1.64023 | 1.65709 |
| N | 1.00906 | 1.25188 | 1.25669 |
| O | 0.71619 | 0.71207 | 0.72253 |
| P | 0.04258 | 0.05035 | 0.05099 |

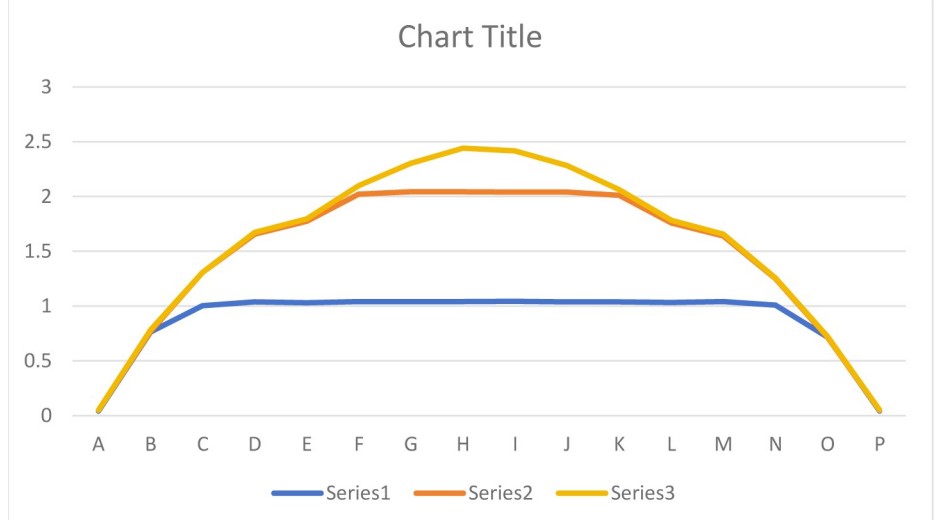

**Fig 4. Peak displacement curve of each unit.**

propagates opposite to the joint plane, and stress attenuation near the joint plane accelerates with increasing distance. However, the rock mass on both sides of the joint plane remains largely unaffected. After the explosion crack connects with the horizontal joint, it significantly influences the subsequent crack propagation.

In Fig 5(B), when there is a joint inclined at 30° between the two gun holes, it substantially alters the rock's damage pattern. During the explosion stress wave stage, both ends of the joints near the gun holes experience the effect of the explosion stress wave. This causes the joints to move toward the back joint plane, leading to the creation of an area with higher crack density at the joint ends, which gradually extends away from the gun holes (250us). This behavior is due to the reflection of the stress wave when it reaches the joint surface, resulting in reflected tensile stress. As the stress wave reaches the joint end, it diffracts and generates a wing crack (350us) at the end. By 550us, the crack continues to expand along the initial crack, the wing crack keeps developing, and the crack at the joint end extends to the far end of the gun hole, connecting with the wing crack. Notably, there are almost no cracks near the hole. This is

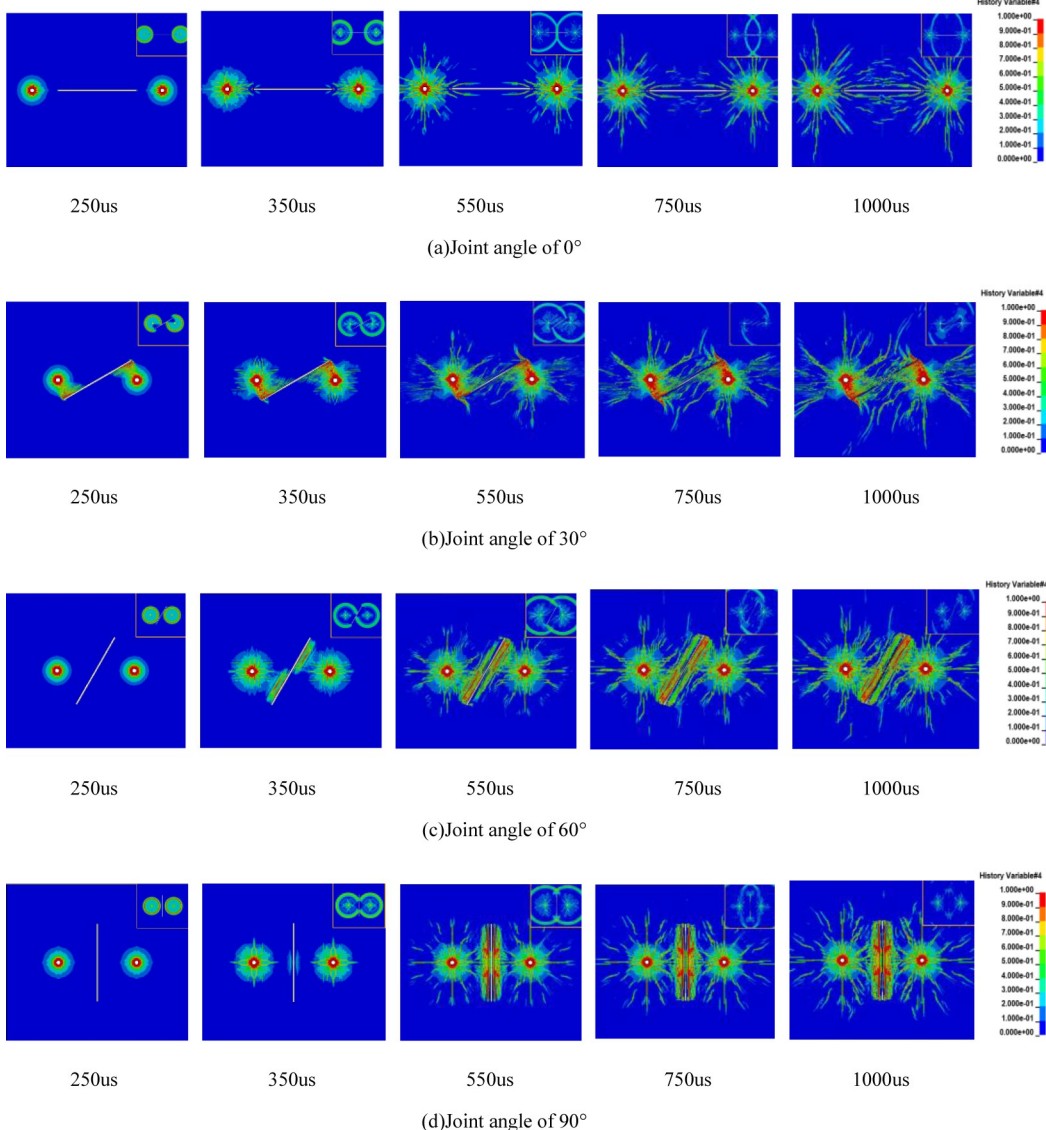

**Fig 5. Evolution of explosive cracks in jointed rock mass without in-situ stress (stress wave cloud image at upper right corner).**

because the presence of the joint hinders the propagation of stress waves, and initial crack formation is impeded. As the stress wave propagates, its gradual attenuation becomes insufficient to cause rock damage (750us).

As illustrated in Fig 5(C), when there is a joint with a 60° inclination angle between the two gun holes, the side of the joint plane near the gun holes initially experiences the effects of compressive stress waves. It forms reflected tensile waves at 350us during the explosion stress wave stage. This stage leads to the formation of a crushed zone and a radial micro-crack zone around the gun holes and generates the main crack. However, there is no local damage occurring at the joint end during this phase. As the stress wave propagates to a greater distance, it gradually weakens. During this time, the explosive gas further extends the original crack on the side away from the joint plane, resulting in a shorter wing crack (550us). Simultaneously, the crack on the side near the joint plane hinders the expansion of the original main crack in

that direction due to the barrier effect of the joint plane. The cracks extend along the direction of the joint surface (750us), leading to the formation of many secondary cracks and resulting in significant damage on both sides of the joint (1000us). Evidently, when the joint angle is 60˚, it has a pronounced promoting effect on crack expansion, resulting in an effective blasting outcome.

In Fig 5(D), when there is a joint with a 90˚ angle between the two gun holes, the joint initially has no impact on crack propagation during the early stage of the explosion stress wave. As the stress wave reaches the joint surface at 350us, rock damage first occurs on both sides of the joint midpoint, forming a crushed zone around the gun holes and radial main cracks. When the stress wave propagates to the joint end, the existing main crack continues to propagate further in its original direction, and reflected tensile stress in the middle of the joint plane causes rock failure (550us). Between 750us and 1000us, there is no significant difference in crack propagation. However, the joint end guides the crack propagation within its region, and the explosive crack tends to incline toward the joint end during its expansion along the original direction. When comparing Figs 2(A) and 5(D), it becomes clear that when the angle between the joint and the line connecting the gun holes is 90˚, the joint does not impede the crack penetration between the gun holes. Instead, it has a promoting effect on crack expansion, leading to a more even distribution of cracks on the joint surface and improved rock fragmentation. Further comparison of the evolution of explosive cracks at different joint angles reveals that the blasting effect is worst at a 30˚ angle. The test results of smooth blasting conducted by Wu Li et al. [34] also indicate that the blasting effect is poorest when the angle between the rock mass's structural plane and the blasting fracture plane is approximately 30˚. The simulation results in this paper align with these test findings.

## 3.3 Analysis of blasting effect of jointed rock mass under different in-situ stress conditions

Fig 6 illustrates the final distribution pattern of explosive crack growth under varying in-situ stress conditions. The key observations from this figure include: When there is no joint between two gun holes (depicted in Fig 6(A)), the growth of explosive cracks is primarily influenced by the in-situ stress condition. Notably, the distribution and length of cracks formed at different lateral stress coefficients ($\lambda$ = 0.5, 1, and 2) are smaller than those without in-situ stress. It's worth highlighting that the vertical crack length is more pronounced when $\lambda$ = 0.5 than under other conditions. The horizontal cracks are suppressed, and vertical crack propagation prevails. Additionally, the cracks tend to incline between the two holes. In cases where $\lambda$ = 1, both horizontal and vertical cracks experience inhibition. The number of secondary cracks is reduced, and the primary cracks are shorter. In contrast to scenarios with $\lambda$ = 0.5, when $\lambda$ = 2, there is a notable promotion of crack formation and propagation in the horizontal direction. More cracks are observed between the holes, and the suppression of primary crack growth in the vertical direction is intensified. In fact, there are virtually no secondary cracks in these conditions. These observations are primarily attributed to radial crack propagation resulting from circumferential tensile stress. When $\lambda$ takes on values of 0.5, 1, and 2, the in-situ stress in all directions surrounding the rock mass is compressive stress, which tends to impede the initiation and propagation of cracks. These findings align with the results presented in the work of Dai Jun and Qian Qihu [35], which focused on parameters related to roadway caving and blasting under high in-situ stress conditions.

In the scenario where the joint angle is 0˚ (refer to Fig 6(B)), and in the absence of in-situ stress, we observe the following phenomena: Explosive cracks propagate in all directions, resulting in a higher number of secondary cracks. The primary cracks in the vertical direction

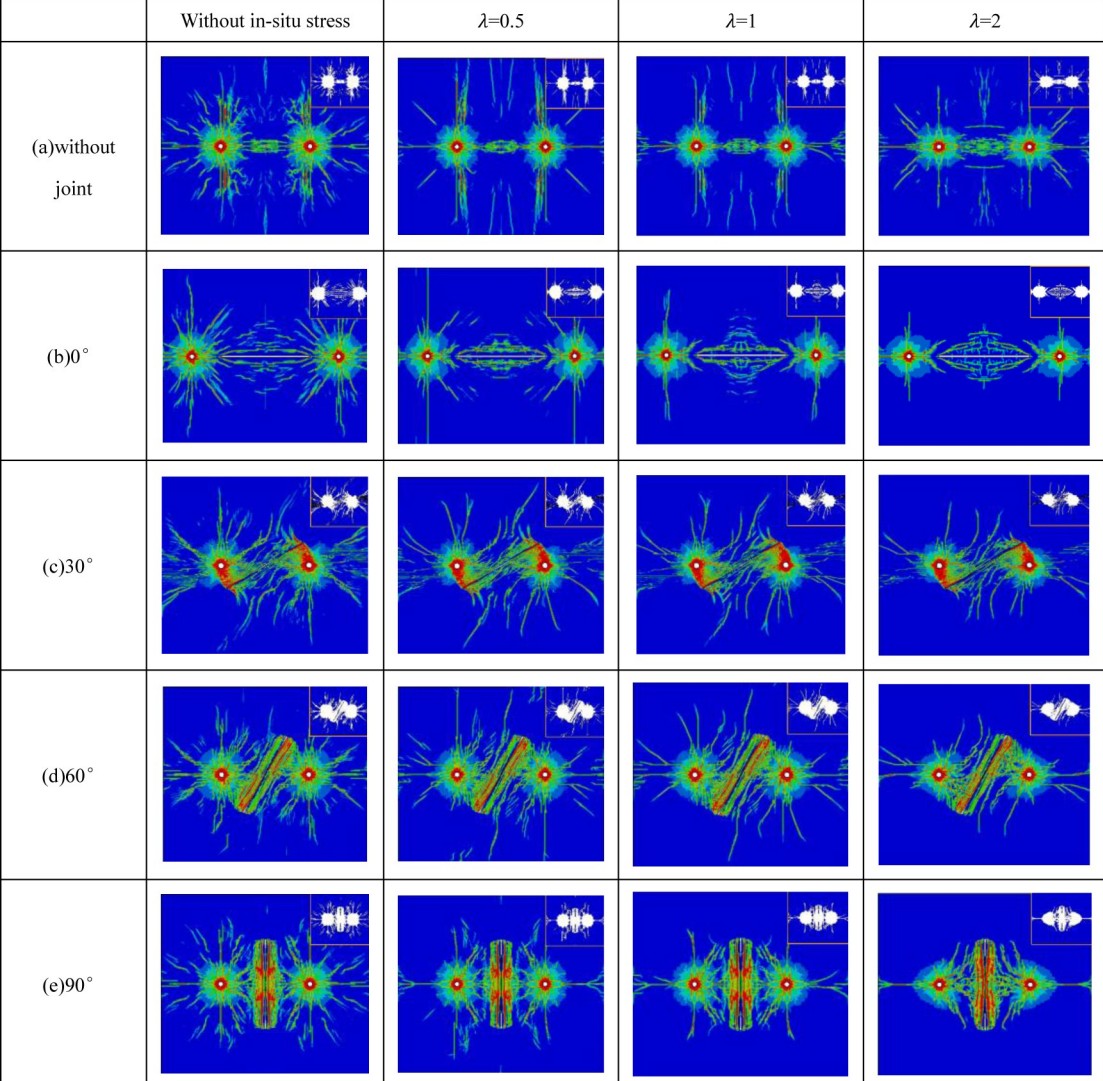

|  | Without in-situ stress | $\lambda=0.5$ | $\lambda=1$ | $\lambda=2$ |
|---|---|---|---|---|
| (a)without joint |  |  |  |  |
| (b)0° |  |  |  |  |
| (c)30° |  |  |  |  |
| (d)60° |  |  |  |  |
| (e)90° |  |  |  |  |

**Fig 6. Distribution diagram of crack growth in rock mass under different in-situ stress conditions (damage range diagram in the upper right corner).**

shift towards the back of the joint due to the influence of stress waves. At a lateral stress coefficient of $\lambda = 0.5$, the vertical cracks extend to the model's boundary, essentially running through the entire model. Horizontal cracks gather on both sides of the joint, with secondary cracks being relatively scarce and primarily concentrated between the two holes. For scenarios with $\lambda = 1$ and $\lambda = 2$, the primary cracks in the vertical direction tend to propagate towards the joint, driven by the effects of in-situ stress. This behavior substantially inhibits the initiation of secondary cracks. The inhibitory effect is more pronounced with $\lambda = 2$, where in-situ stress plays a stronger role in suppressing crack growth. As a result, the initiation and propagation of secondary cracks are significantly hindered under conditions of in-situ stress. Now, when the joint angle is 30° (as shown in Fig 6(C)), the explosive crack extends towards the joint's end, forming a wing crack on the back joint plane during the propagation process. This crack extends away from the gun hole on the side closest to the gun hole and intersects with the wing crack generated by another gun hole. While the initial in-situ stress still has some inhibitory

effect on secondary crack initiation and propagation, this effect is less pronounced than for other joint angles. Notably, there are no cracks observed at both ends of the joint, which is attributed to stress concentration within the joint. In summary, the presence of initial in-situ stress and a joint angle of 30° results in the joint playing a prominent role in guiding and promoting explosive crack propagation. Under these conditions, the joint primarily controls the direction of crack propagation.

In the case where the joint angle is 60° (as depicted in Fig 6(D)), we observe the following behaviors: Secondary cracks are inhibited, and the primary crack expands towards the direction of the maximum principal stress. The rock damage on the joint plane closest to the gun hole is more extensive, and the cracks are predominantly oriented towards both ends of the joint. Under conditions of lateral stress coefficients $\lambda = 0.5$ and $\lambda = 1$, a long primary crack originates from the joint. These cracks almost traverse the entire model, although the suppression of primary cracks at the hole varies. With $\lambda = 2$, it is evident that the inhibitory effect on crack propagation in the vertical direction is more pronounced, favoring horizontal crack growth. When the joint angle is 90° (as shown in Fig 6(D)), for lateral stress coefficients $\lambda = 0.5$ and $\lambda = 1$, crack propagation on both sides of the joint tends to incline towards the direction of the gun hole. However, with $\lambda = 2$, the cracks extend towards the joint's end, and the joint exhibits a clear guiding effect on the explosive cracks. This phenomenon results from the combined influence of joint stress and joint guidance, with the direction of crack propagation being influenced by both the joint and the initial in-situ stress. To summarize, the specific joint angle and lateral stress coefficients significantly affect the patterns of crack propagation, with different conditions leading to variations in crack distribution, orientation, and the extent of primary and secondary cracks in the rock mass. These observations highlight the complex interplay of geological and stress factors in controlling the blasting process.

Fig 7 presents the changes in the area of the blasting damage zone under different in-situ stress conditions and joint angles. When the lateral stress coefficient $\lambda$ remains constant and the burial depth is increased, variations in blasting effectiveness are observed. To facilitate meaningful comparisons, this study incorporates diverse in-situ stress values for a consistent lateral stress coefficient. It also considers the absence of a joint as the baseline for the damage area when the joint angle is 0°, although it should be noted that this specific scenario deviates from other blasting models in terms of hole positions. As depicted in Fig 6, it becomes evident that maintaining a fixed lateral stress coefficient $\lambda$ while elevating the vertical in-situ stress from 10MPa to 20MPa results in decreased crack expansion length and a reduced damage area. Clearly, as the burial depth increases and the in-situ stress intensifies, the initiation and expansion of cracks become more restricted. This conclusion aligns with the findings reported by Yang Jianhua and their colleagues [36]. In summary, the interaction of varying in-situ stress and joint angles has a pronounced impact on the extent of damage caused by blasting. The study reveals the complex relationship between these factors and provides valuable insights into optimizing blasting procedures in different geological conditions.

Fig 7 presents the area of the blasting damage zone under various conditions, taking into account vertical in-situ stress $\sigma_v$ at 10MPa and 20MPa, lateral stress coefficients at 0.5, 1, and 2, and joint angles at 30°, 60°, and 90°. Notably, when holding the lateral stress coefficient constant and varying the joint angles, the damage zone area tends to increase with an increasing joint angle. However, it's crucial to recognize that the damage zone area doesn't exhibit indefinite growth with larger angles; instead, there appears to be an optimal value. In this study, the most extensive blasting damage area within the rock mass was observed when the joint angle was set at 60° and $\sigma_v$ was 10MPa. Additionally, jointed rock masses display larger damage areas compared to unjointed ones, and the presence of a joint results in the smallest damage area when the joint angle is 30°. In conclusion, the research sheds light on the interplay

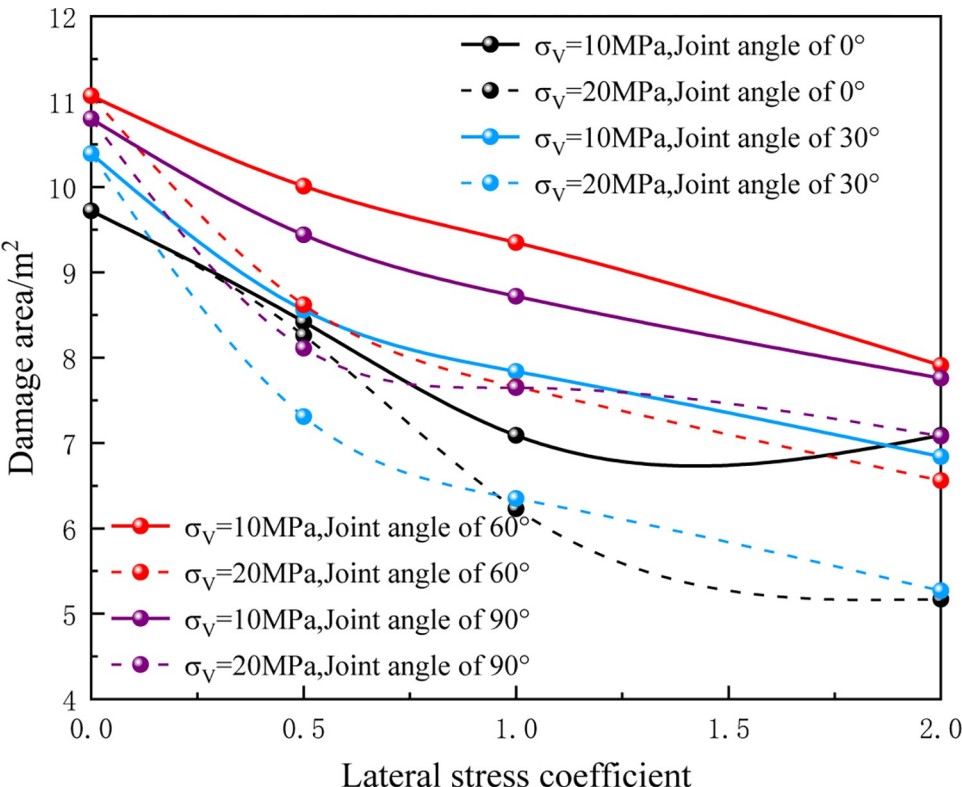

**Fig 7. The area of blasting damage zone formed under different conditions.**

between in-situ stress conditions, joint angles, and blasting damage. These findings provide valuable insights into the optimization of blasting practices across various geological contexts.

## 4 Conclusion

In this study, numerical simulations of the blasting process in jointed rock masses under various conditions were conducted, yielding the following main conclusions:

1. When an open joint is present in the rock mass, and the joint thickness doesn't align with the joint plane, maximum element displacement occurs at the joint plane. Stress concentration is observed at both ends of the joint, while the stress between the two ends is relatively mild, resulting in a larger damage area. When the joint thickness is too small, maximum stress occurs at the center of the hole connection. The presence of a joint exerts a significant guiding effect on crack propagation.

2. The existence of a joint imposes a barrier to crack expansion and penetration. When the explosion stress wave reaches the joint surface, it hinders the propagation of the stress wave, leading to the diffusion of wing cracks at the joint end. Under similar lateral stress coefficients and joint angles, the area of the blasting damage zone decreases with increasing initial in-situ stress.

3. Under the same initial in-situ stress conditions, the area of the blasting damage zone initially increases and then decreases as the joint angle increases. However, it remains larger than that in the absence of a joint, and there is an optimal angle that maximizes the damage

area. In simulated conditions, the largest damaged crack area is observed when the joint angle is 60°.

4. The presence of initial in-situ stress has a certain impact on the initiation and expansion of blasting cracks. The degree and pattern of this influence are not only related to the lateral stress coefficient but also associated with the joint's angle and thickness. When in-situ stress is present, the pressure effect from the initial in-situ stress field doesn't favor the initiation and propagation of blasting cracks. Nevertheless, the presence of joints exhibits a clear guiding and promoting effect on crack propagation. The morphology of crack propagation is jointly controlled by both the joints and in-situ stress conditions.

## Author Contributions

**Data curation:** Chen Cao.

**Formal analysis:** Chen Cao.

**Investigation:** Yadi Wang, Mingda Li.

**Supervision:** Hai Rong, Yadi Wang.

**Validation:** Hai Rong, Jincheng Li.

**Writing – original draft:** Nannan Li.

**Writing – review & editing:** Hai Rong.

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
