## [Decision Letter · Decision Letter 0]

2 Jan 2024

PONE-D-23-39104Numerical simulation of rock blasting under different

in-situ stresses and joint conditionsPLOS ONE

Dear Dr. Li,

Thank you for submitting your manuscript to PLOS ONE. After careful consideration, we feel that it has merit but does not fully meet PLOS ONE’s publication criteria as it currently stands. Therefore, we invite you to submit a revised version of the manuscript that addresses the points raised during the review process.

We look forward to receiving your revised manuscript.

Kind regards,

Dr. S. M. Anas, Ph.D.(Structural Engg.), M.Tech(Earthquake Engg.)

Academic Editor

PLOS ONE

Journal Requirements:

"This work was supported by the National Natural Science Foundation of China Project No.51904145, Basic scientific research project (youth project) of Liaoning Provincial Department of Education in 2022 No. LJKQZ20222322, the Engineering Laboratory of Deep Mine Rockburst Disaster Assessment Open Project No. LMYK2020006, the Liaoning Natural Science Foundation Program Guidance Plan No. 2019-ZD-0045, and the Liaoning Provincial Department of Education Project No. LJ2019JL007."

Additional Editor Comments:

Dear Authors:

The manuscript titled "Numerical simulation of rock blasting under different in-situ stresses and joint conditions" [PONE-D-23-39104] was submitted for peer review to two experts (peer reviewers) in the relevant research field. One reviewer suggested “Minor Revision” and the other recommended "Major Revision" and provided insightful comments regarding the research novelty, numerical verification, modeling aspects, insufficient literature and general layout of the manuscript. After considering the reviewers' recommendations and conducting a preliminary analysis of the paper, this editor has decided to proceed with a "Major Revision" for this submission.

Important note from this academic editor, Dr. S. M. Anas: -

I would like to bring to your attention that citing the papers suggested by the reviewers is not mandatory for your revised manuscript. It is entirely up to you whether or not you choose to include the suggested papers in your revised version. The reviewers have provided these suggestions to enhance the quality and credibility of your research, but ultimately, the decision is yours. You have the freedom to decline including any of the suggested papers in your revised manuscript if you feel they are not relevant or do not add value to your study.

I look forward to receiving your revised version of the manuscript.

Thank you for your time.

Sincerely yours,

Dr. S. M. Anas

(Academic Editor)

Reviewers' comments:

Reviewer's Responses to Questions

**Comments to the Author**

1. Is the manuscript technically sound, and do the data support the conclusions?

Reviewer #1: Yes

Reviewer #2: Yes

2. Has the statistical analysis been performed appropriately and rigorously? 

Reviewer #1: N/A

Reviewer #2: Yes

3. Have the authors made all data underlying the findings in their manuscript fully available?

Reviewer #1: No

Reviewer #2: Yes

4. Is the manuscript presented in an intelligible fashion and written in standard English?

Reviewer #1: Yes

Reviewer #2: Yes

5. Review Comments to the Author

Reviewer #1: After careful analysis, the following concerns and questions arose:

1) the summary lacks emphasis on aspects of novelty

2) the introduction chapter should be numbered 1, not 0,

3) the literature review is not complete, a number of works aimed at the study of rock properties, the choice of constitutive models, or how to describe the process of interaction of, for example, the blast wave with the rock mass are missing, I encourage you to study some of the following works: DOI:10.1016/j.jrmge.2021.12.022 , DOI:10.1016/j.ijimpeng.2022.104484, DOI:10.1016/j.ijimpeng.2020.103543, DOI:10.1016/j.jrmge.2020.09.007 or DOI:10.1016/j.ijmecsci.2022.107197.

4) No consideration of residual stress state. I encourage you to study the paper: DOI:10.1155/2019/2878969.

5) There is no conversion of the process of interaction of the blast wave with the rock (section) and the effects associated with it - multiple reflections of the wave in the hole and overlapping of pressure waves or vacuum effect.

6) A key aspect is missing, namely, an examination of the sensitivity of the model to initial-boundary conditions and the density of the FEM mesh. A key parameter in describing the destruction/cracking process of rock material.

7) There is no consideration of the heterogeneity of the rock material by taking into account cracks, defects or inclusions.

8) There is no reference to experimental testing - no description of the validation process.

9) There is no analysis in terms of the sequence of ignition initiation - the sequence of explosions - delays and what effects this has.

Reviewer #2: - The overall findings of this paper are commendable and align well with previous research conducted in this field.

- The explanation of the obtained results was very comprehensive.

- However, it is recommended to make amendments to the abstract by providing more detailed information on the methodology while minimizing the emphasis on findings. The conclusions already provide a comprehensive overview of the results.

- Improve Figure 1 (looks overlapping)

- Include the rock types used for simulation

- Edit Figure 6 (Lambda)

6. PLOS authors have the option to publish the peer review history of their article (what does this mean?). If published, this will include your full peer review and any attached files.

Reviewer #1: No

Reviewer #2: No

---

## [Author Response · Author response to Decision Letter 0]

8 Jan 2024

Reply to the first reviewer's comments

Dear reviewer,

Thank you very much for reviewing our paper, and providing comments and suggestions to us. I am Nannan Li, the corresponding author of the paper “Numerical simulation of rock blasting under different in-situ stresses and joint conditions (PONE-D-23-39104)”

I have revised my manuscript according to the changes requested by the reviewer, and now I will reply to the comments item by item. Thank you！

Comment 1: 

The summary lacks emphasis on aspects of novelty.

Answer 1:

Thank the reviewer for his valuable comments.

You correctly point out the lack of emphasis on novelty in the abstract. The following is a revised partial abstract, highlighting the novelty of the study:

“... Numerical simulation model. In this study, the effects of lateral stress coefficient, joint width, joint Angle and other parameters on crack generation and propagation of rock mass are studied by finite element numerical simulation. Through stress analysis and damage area comparison, the relationship between damage crack growth and horizontal and vertical stress difference of rock mass is explored, and the action principle of these factors is analyzed. ...”

Thanks to the reviewers.

Comment 2: 

The introduction chapter should be numbered 1, not 0.

Answer 2:

Thank the reviewer for his valuable comments.

The author has revised the Revised Manuscript with Track Changes according to your comments.

Thanks to the reviewers.

Comment 3: 

The literature review is not complete, a number of works aimed at the study of rock properties, the choice of constitutive models, or how to describe the process of interaction of, for example, the blast wave with the rock mass are missing, I encourage you to study some of the following works: DOI:10.1016/j.jrmge.2021.12.022,DOI:10.1016/j.ijimpeng.2022.104484,

DOI:10.1016/j.ijimpeng.2020.103543,

DOI:10.1016/j.jrmge.2020.09.007orDOI:10.1016/j.ijmecsci.2022.107197.

Answer 3:

Thank you very much for your comments and recommendations of relevant literature.

Thanks to the reviewers for the relevant research papers. By studying these literatures, I have a more comprehensive understanding of the study of rock properties, the selection of constitutive models and the interaction between explosion waves and rock mass. These literatures provide important reference and guidance for my research. I have quoted these documents in the revised draft to better supplement the completeness of the literature review in the manuscript, and marked them in blue font.

Thanks to the reviewers.

Comment 4: 

No consideration of residual stress state. I encourage you to study the paper: DOI:10.1155/2019/2878969.

Answer 4:

Thank the reviewer for his valuable comments.

You mentioned that the effect of the residual stress state was not considered, and I have read the paper you recommended and learned more about the research in this area. In my research, the state of residual stress is not the focus of the research, but the influence of ground stress on the development and propagation of cracks in jointed rock mass is mainly considered.

Through reading this paper, I realized the influence of the state of residual stress on the behavior of rock mass. In order to improve and deepen my research, I plan to explore more deeply the influence of residual stress on the materials or systems under study in future work. I will also consider more comprehensively the effects of various stress states on the behavior of materials in order to provide a more accurate and complete analysis. At the same time, I will quote the paper you recommended in my research to fully reflect the influence of the residual stress state on the research and ensure that my research results are more complete and accurate.

Thank you very much for the reviewer's guidance and the recommendation of relevant literature.

Comment 5: 

There is no conversion of the process of interaction of the blast wave with the rock (section) and the effects associated with it - multiple reflections of the wave in the hole and overlapping of pressure waves or vacuum effect.

Answer 5:

Thank you very much for your comments.

In my research, I mainly study the interaction process between explosion shock wave and rock from a macroscopic perspective. In the research process, the interaction is mainly reflected in crack initiation, expansion and stress wave dissipation of rock mass. Secondly, the influences of the width, Angle of joint and lateral stress coefficient on crack propagation of rock mass are discussed.

Your guidance is very important, including the multiple reflections of shock waves in rock pores and the complexity of the propagation path, which is not viewed from a microscopic perspective in my research content. In the future research, I will further explore the physical process of shock wave propagation and rock interaction, including the multiple reflections of waves in holes and the propagation path of waves. This will contribute to a more comprehensive understanding of the impact of shock waves on rocks and improve the scientific research.

Thanks to the reviewers.

Comment 6:

A key aspect is missing, namely, an examination of the sensitivity of the model to initial-boundary conditions and the density of the FEM mesh. A key parameter in describing the destruction/cracking process of rock material.

Answer 6:

First of all, thank you very much for your comments. You correctly pointed out that my research lacks the sensitivity of initial boundary conditions and the sensitivity of finite element mesh density. And, you also mentioned that I need to describe the key parameters of the rock material failure/cracking process in the study. Here are the responses to your suggestions:

First of all, in this study, I set the initial boundary conditions, set the boundary of the model as non-reflective boundary conditions, simulate the infinite domain of the rock mass, and set the normal constraint in the Z direction. Secondly, the grid Quality Check is to evaluate the grid quality through the LS-PrePost internal function, click EleTol - EleEdit, and then select Check - Quality Check. There are many indicators to evaluate the quality of the grid, including Aspect Ration, Jacobian, Warapage and so on. The following is a model as an example, solid unit volume inspection, free solid unit inspection, repeated solid unit inspection, etc. Then, in the simulation process, the dynamic tensile strength, dynamic compressive strength and plastic strain of rock are mainly used as the key parameters of rock material failure/cracking.

In future studies, I will conduct more sensitivity analyses to assess the model's response to initial boundary conditions. I will study the effect of these conditions on the failure and cracking process of rock mass by varying different combinations of the initial boundary conditions, such as the initial stress and strain conditions. This will help me understand changes in the behavior of rock materials and the stability of models.

Thanks to the reviewers.

Comment 7:

There is no consideration of the heterogeneity of the rock material by taking into account cracks, defects or inclusions.

Answer 7:

Thank you for your valuable comments. Your comments are very valuable.

First of all, the heterogeneity of rock materials has a certain impact on crack propagation and the magnitude of ground stress. However, this study mainly takes homogeneous rock mass as the research object. In local rock mass, it is considered to be isotropic and has a kind of rock property, such as granite. In future studies, I plan to introduce more data and models on rock heterogeneity to more fully analyze the effect of ground stress on crack propagation.

Secondly, the cracks, defects or inclusions you mentioned are another important factor, which can also have a significant effect on the fracture and crack growth of the rock. The existence of these factors may change the mode of action of earth stress. In this paper, I think that the open joint can be regarded as one of the cracks and defects in rock mass. However, the combination of cracks and defects is very different, and my consideration is not comprehensive. At present, our consideration focuses on starting from the homogeneous rock mass, thoroughly studying its composition and other aspects, and then using the control variable method to add cracks, defects or inclusions one by one to turn them into a combination of parameters. This is the content of my next research, so as to more accurately evaluate the influence of different parameter combinations on rock behavior.

Thanks to the reviewers.

Comment 8:

There is no reference to experimental testing - no description of the validation process.

Answer 8:

Thank you for your valuable comments. Your comments are very valuable.

You pointed out that I did not mention the validation process of experimental tests in the study, and I recognize that this is an area that needs to be added to enhance the reliability of the study. However, there are many factors to consider in the existing laboratory tests, such as the limitation of confining pressure applied by rock mass and the harm of explosion wave to the experimenter. In the future research work, I will actively consider conducting experimental tests and describing experimental equipment, methods and results in detail to verify the accuracy and reliability of numerical simulation results.

Thanks to the reviewers.

Comment 9:

There is no analysis in terms of the sequence of ignition initiation - the sequence of explosions delays and what effects this has.

Answer 9:

Thank the reviewer for his valuable comments.

First of all, the sequence of ignition has an important effect on the explosion effect and result. It is a limitation of the study to analyze the sequence of initiation points. In this study, initiation mode is not taken as the focus of this research. In order to simplify the model, a quasi-two-dimensional model is adopted in the research process, and the initiation point is not defined, and the whole initiation mode is adopted. In the following research, 3D modeling will be used, and the initiation point will be defined, different initiation sequences will be considered and their impact on the results will be analyzed. This will lead to a better understanding of the explosion sequence and the effect of explosion wave effects on rock behavior.

Secondly, explosion delay is also an important factor in the study. In the current study, I did not consider the timing and delay of the explosion. In future studies, I will further study the timing and delay of the explosion and analyze its impact on the results to provide a more comprehensive and accurate analysis of the blasting effect.

Thanks to the reviewers.

Reply to the second reviewer's comments

Comment 1: 

It is recommended to make amendments to the abstract by providing more detailed information on the methodology while minimizing the emphasis on findings. The conclusions already provide a comprehensive overview of the results.

Answer 1:

Thank the reviewer for his valuable comments.

In the revised draft, I have revised it according to the teacher's comments and provided more detailed information about the method. The revised content has been marked in blue font.

Thanks to the reviewers.

Comment 2: 

Improve Figure 1 (looks overlapping)

Answer 2:

Thank the reviewer for their valuable comments.

I have revised Figure 1 while revising the manuscript and track changes.

Thanks to the reviewers.

Comment 3: 

The rock types used for simulation.

Answer 3:

Thank the reviewer for their valuable comments.

In 2.3, I modified the relevant part and marked it with blue font.

Thanks to the reviewers.

Comment 4: 

Edit Figure 6 (Lambda)

Answer 4:

Thank the reviewer for their valuable comments.

I have revised Figure 6 while revising the manuscript and track changes.

Thanks to the reviewers.

Thank you for giving me the chance to revise our manuscript!

Best regards, 

Nannan Li

Liaoning Technical University, China.

---

## [Decision Letter · Decision Letter 1]

16 Jan 2024

PONE-D-23-39104R1Numerical simulation of rock blasting under different in-situ stresses and joint conditionPLOS ONE

Dear Dr. Li,

Thank you for submitting your manuscript to PLOS ONE. After careful consideration, we feel that it has merit but does not fully meet PLOS ONE’s publication criteria as it currently stands. Therefore, we invite you to submit a revised version of the manuscript that addresses the points raised during the review process.

We look forward to receiving your revised manuscript.

Kind regards,

Dr. S. M. Anas, Ph.D.(Structural Engg.), M.Tech(Earthquake Engg.)

Academic Editor

PLOS ONE

Journal Requirements:

Additional Editor Comments:

Dear Authors,

I trust this email finds you well. I am writing to update you on the status of your manuscript entitled "Numerical simulation of rock blasting under different in-situ stresses and joint conditions" (PONE-D-23-39104R1) following the recent rounds of review.

I am pleased to inform you that both of the previous reviewers have recommended acceptance of the revised manuscript. Reviewer 2, in particular, commended the revisions made but suggested a very minor comment regarding the need for additional information on the construction of the numerical model, if available.

Upon conducting a preliminary analysis myself, I have observed that the references suggested by Reviewer 1 have been incorporated into the revised manuscript. However, it is crucial to ensure that these references are relevant to your study. If they do not contribute significantly, I kindly request that you consider removing them at this stage of the revision.

If, on the other hand, you believe that the suggested references are pertinent to your work, I request that you provide a valid reason for their inclusion in your response.

Your prompt attention to these matters is appreciated, as the final decision on the manuscript will be influenced by your responses to the above queries. Please address the comment from Reviewer 2 regarding additional information on the numerical model and the relevance of the cited references suggested by Reviewer 1.

Thank you for your cooperation and diligence throughout this process. I look forward to receiving your response.

Best regards,

Dr. S. M. Anas

Reviewers' comments:

Reviewer's Responses to Questions

**Comments to the Author**

1. If the authors have adequately addressed your comments raised in a previous round of review and you feel that this manuscript is now acceptable for publication, you may indicate that here to bypass the “Comments to the Author” section, enter your conflict of interest statement in the “Confidential to Editor” section, and submit your "Accept" recommendation.

Reviewer #1: All comments have been addressed

Reviewer #2: All comments have been addressed

2. Is the manuscript technically sound, and do the data support the conclusions?

Reviewer #1: Yes

Reviewer #2: Yes

3. Has the statistical analysis been performed appropriately and rigorously? 

Reviewer #1: N/A

Reviewer #2: Yes

4. Have the authors made all data underlying the findings in their manuscript fully available?

Reviewer #1: Yes

Reviewer #2: Yes

5. Is the manuscript presented in an intelligible fashion and written in standard English?

Reviewer #1: Yes

Reviewer #2: Yes

6. Review Comments to the Author

Reviewer #1: The authors have made a lot of explanations at this stage. Selected issues have been included in the new version of the work. At this stage, I recommend the article for further publishing processes.

Reviewer #2: All comments have been well addressed. The detailed and comprehensible explanation of the results and analysis is appreciated. It is recommended to provide additional information on the construction of the numerical model (If any), ensuring that all details are thoroughly explained. This will enable readers to easily comprehend and cite this paper in the future. Well done

7. PLOS authors have the option to publish the peer review history of their article (what does this mean?). If published, this will include your full peer review and any attached files.

Reviewer #1: No

Reviewer #2: No

---

## [Author Response · Author response to Decision Letter 1]

4 Feb 2024

Many thanks to the review teachers and the editorial department for providing valuable suggestions for revision. Here are the answers to the questions:

Reviewer 1：additional information on the numerical model

Thank the reviewer for his valuable comments.

Complete digital model information is provided in the manuscript.

Thanks to the reviewers.

Reviewer 2：the relevance of the cited references

DOI: 10.1016 / j.jmecsci.2022.107197, this paper is different from the constitutive model calculation method of the manuscript, but both are useful for the study of damage parameters and cracks, which is related to the manuscript.

DOI: 10.1016 / j.jimpeng.2020.103543, this paper introduced the determination method of the parameters of the dolomite JH-2 model, which is conducive to future research and reference.

DOI: 10.1016 / j.jimpeng.2022.104484, this paper evaluated the performance of the three models under different stress conditions, and the manuscript has some similarities with them, which is relevant and conducive to further research.

DOI: 10.1016 / j.jrmge.2021.12.022, the study on the effects of heterogeneity and initial cracks on failure and cracking modes in this paper is related to the content of the manuscript, and both have initial cracks, which is very valuable for reference research.

DOI:10.1155/2019/2878969, this paper is a multi-scale modeling of rock mass and lays the foundation for the initial conditions of the three-dimensional finite element model. The initial conditions of rock mass are also considered in the manuscript.

To sum up, I think the added references are related to the manuscript, so quoting the papers proposed by the reviewer is conducive to further research.

Thanks to the reviewers.

During self-examination, it was found that the original references 33 and 35 were repeated in the Manuscript, and the revisions were made in the reply Revised Manuscript with Track Changes 2.

---

## [Editor Report · Decision Letter 2]

7 Feb 2024

Numerical simulation of rock blasting under different in-situ stresses and joint condition

PONE-D-23-39104R2

Dear Dr. Li,

We’re pleased to inform you that your manuscript has been judged scientifically suitable for publication and will be formally accepted for publication once it meets all outstanding technical requirements.

Kind regards,

Dr. S. M. Anas, Ph.D.(Structural Engg.), M.Tech(Earthquake Engg.)

Academic Editor

PLOS ONE

Additional Editor Comments (optional):

Dear Authors,

I hope this email finds you well. I am writing to inform you about the outcome of the reevaluation of your manuscript titled "Numerical simulation of rock blasting under different in-situ stresses and joint condition" [PONE-D-23-39104R2] following the revision.

Upon careful reevaluation, I am pleased to note that you have satisfactorily addressed the comments raised during the review process, particularly regarding the justification for including the references suggested by the reviewers. It is evident that you have carefully considered the relevance of these references to your study, as suggested in my previous correspondence: "Upon conducting a preliminary analysis myself, I have observed that the references suggested by Reviewer 1 have been incorporated into the revised manuscript. However, it is crucial to ensure that these references are relevant to your study. If they do not contribute significantly, I kindly request that you consider removing them at this stage of the revision."

Based on my assessment, I am inclined to recommend an acceptance decision for your revised manuscript, pending the approval of the editorial board.

Please note that this recommendation is subject to the final approval of the editorial board of PLOS ONE. Once their approval is obtained, you will receive further instructions regarding the next steps in the publication process.

I commend you on the thorough revisions made to your manuscript and appreciate your diligence in addressing the reviewers' comments. Should you have any questions or require further clarification, please do not hesitate to contact me.

Thank you for choosing PLOS ONE as the outlet for your research, and I look forward to the successful publication of your work.

Best regards,

Dr. S. M. Anas

Academic Editor

PLOS ONE
---

## [Editor Report · Acceptance letter]

1 Mar 2024

PONE-D-23-39104R2 

PLOS ONE

Dear Dr. Li, 

I'm pleased to inform you that your manuscript has been deemed suitable for publication in PLOS ONE. Congratulations! Your manuscript is now being handed over to our production team.

Kind regards, 

on behalf of

Dr. S. M. Anas 

Academic Editor

PLOS ONE